# The airglow layer emission altitude cannot be determined unambiguously from temperature comparison with lidars

**Tim Dunker**[1,*]

[1]Department of Physics and Technology, UiT The Arctic University of Norway, Postboks 6050 Langnes, Tromsø, Norway
[*]now at: National laboratory, Justervesenet, Postboks 170, 2027 Kjeller, Norway

**Correspondence:** T. Dunker (tdu@justervesenet.no)

**Abstract.** I investigate the nightly mean emission height and width of the OH*(3–1) layer by comparing nightly mean temperatures measured by the ground–based spectrometer GRIPS 9 and the Na lidar at ALOMAR. The data set contains 42 coincident measurements between November 2010 and February 2014, when GRIPS 9 was in operation at the ALOMAR observatory (69.3°N, 16.0°E) in northern Norway. To closely resemble the mean temperature measured by GRIPS 9, I weight each nightly mean temperature profile measured by the lidar using Gaussian distributions with 40 different centre altitudes and 40 different full widths at half maximum. In principle, one can thus determine the altitude and width of an airglow layer by finding the minimum temperature difference between the two instruments. On most nights, several combinations of centre altitude and width yield a temperature difference of $\pm 2$ K. The generally assumed altitude of 87 km and width of 8 km is never an unambiguous, good solution for any of the measurements. Even for a fixed width of $\sim$8.4 km, one can sometimes find several centre altitudes that yield equally good temperature agreement. Weighted temperatures measured by lidar are not suitable to determine unambiguously the emission height and width of an airglow layer. However, when actual altitude and width data are lacking, a comparison with lidars can provide an estimate of how representative a measured rotational temperature is of an assumed altitude and width. I found the rotational temperature to represent the temperature at the commonly assumed altitude of 87.4 km and width of 8.4 km to within $\pm$16K, on average. This is not a measurement uncertainty.

## 1 Introduction

To evaluate whether the common assumption of a nightly mean hydroxyl (OH) layer emission altitude of 87 km and a fixed width of 8 km is justified, I compare temperatures measured by the Na lidar at ALOMAR with OH*(3–1) rotational temperature measured by GRIPS 9. Both instruments were located at the ALOMAR observatory (69°N) in Norway between November 2010 and February 2014, resulting in 42 coincident measurements. To determine the emission altitude and width of the OH*(3–1) layer, I compute Gaussian–weighted mean lidar temperatures for different centre altitudes and widths to resemble the temperatures measured by GRIPS 9. In principle, the minimum of the temperature difference then yields an estimate of the emission altitude and width of the OH*(3–1) layer.

The emission altitude and width of the mesospheric OH* layer are essential quantities not only for the determination of temperature trends (e.g. Espy and Stegman, 2002; Beig, 2011), but also for the comparison with temperature measurements by meteor radars, satellites, and resonance lidars. Typically, studies of ground–based infrared observations assume a stationary emission altitude of 87 km and a full width at half maximum (FWHM) of about 8 km, which correspond roughly to recommended values given by Baker and Stair Jr. (1988).

Apart from satellite–based observations and the direct measurement by rockets, the OH* emission layer height has been estimated by comparing ground–based measurements of the OH* rotational temperature with temperature measurements by lidars. From such a comparison of three nights of data, von Zahn et al. (1987) found an emission layer altitude of (86±4) km. The FWHM of Gaussian function was fixed at 8.4 km (von Zahn et al., 1987).

Pautet et al. (2014, Fig. 4) compared nightly mean temperatures of the Advanced Mesospheric Temperature Mapper to those measured by an Na lidar located in Logan, Utah. Their results show that, for a Gaussian–shaped weighting function centred at 87 km and a full width at half maximum of 9.3 km, the Na lidar temperatures are apparently warmer than those measured by the OH* imager, on average (Pautet et al., 2014). The centre altitude was not varied in these studies, and lidar temperatures appear to be warmer by roughly 10 K, on average (Pautet et al., 2014), while the difference exceeds 20 K on certain days (Pautet et al., 2014). Such a temperature difference can in principle be a systematic measurement error. By varying the centre altitude and the FWHM of the applied Gaussian function, I clarify whether such a temperature difference between the Na lidar and GRIPS 9 can also arise because of the choice of parameters. From the temperature difference at each day, I estimate how representative the OH*(3–1) rotational temperature is of the temperature at 87 km (She and Lowe, 1998), assuming a stationary width of 8.4 km.

## 2   Instruments and methods

I use temperature measurements made by the Ground–based Infrared P–branch Spectrometer (GRIPS) 9, which probes the OH*(3–1) vibrational transition, and the Na lidar at ALOMAR. The ALOMAR observatory is located at 69.3°N, 16.0°E. The time period covered by GRIPS 9 at ALOMAR extends from October 2010 to May 2014. For this study, I analyze lidar data from measurements in darkness only, because GRIPS can only observe the OH* nightglow. This means that there is hardly any or no data from GRIPS 9 between May and August. Lidar observations are limited to clear nights. The data set consists of 42 coincident measurements, with both instruments' nightly temperature time series restricted to the same periods of time. The nightly mean temperatures computed from GRIPS 9 are based on measurements with a temporal resolution of one minute.

The GRIPS instruments were described by Schmidt et al. (2013). GRIPS uses an InGaAs array detector to measure the airglow spectrum between approximately 1522 nm and 1545 nm. Rotational temperatures are derived from the OH*(3–1) P₁–branch. The rotational lines of this vibrational level are less affected by non–local thermodynamic equilibrium effects compared to higher vibrational levels (Noll et al., 2015). The analysis of GRIPS data is described in Schmidt et al. (2013, Sect. 3.3). One detail is worth mentioning: the derived temperature is sensitive to the Einstein coefficients used in the analysis (e.g., Noll et al., 2015). To derive temperatures from GRIPS data, Einstein coefficients published by Mies (1974) are used. Using Einstein coefficients published by Langhoff et al. (1986) instead of those by Mies (1974) leads to apparent OH*(3–1) temperatures colder by $(3.5 \pm 0.3)$ K, on average (C. Schmidt, pers. comm.).

The two instruments used in this analysis were co–located, but their fields of view differed considerably. The Na lidar has a field of view of $600\ \mu$rad, which corresponds to $9{\cdot}10^{-3}$ km² at an altitude of 87 km. The nominal field of view of GRIPS derived from the F–number of the spectrograph is $15°$ (Schmidt et al., 2013). A laboratory assessment of the field of view of GRIPS revealed that the effective acceptance angle is slightly smaller ($\sim 14°$ instead of $15°$). Nevertheless, the field of view of GRIPS 9 is larger than $400$ km² at 87 km. Although the fields of view overlap or are close to each other (depending on the lidar zenith angle), one cannot expect the measured temperatures to be exactly equal. Due to waves and other processes, atmospheric temperature varies across the field of view of GRIPS on various scales, and the lidar probes only a small part of this volume.

Each of the lidar's two beams has 300 altitude channels with a time resolution of 1 μs, yielding a nominal altitude resolution of 150 m in the zenith direction. Therefore, a temperature profile consists of many independent measurements. The difference in measured temperature by the lidar's two beams is an estimate of the horizontal temperature variability. Usually, one beam points to the north at a zenith angle of 20°, and the other beam to the east at the same zenith angle. At an altitude of 92 km, the horizontal distance between the two beams is approximately 50 km. For each of the 42 nights, I choose lidar data from the beam with the smallest temperature uncertainty, and only temperatures with an uncertainty smaller than or equal 5 K. Whether one computes the mean temperature profile from only one beam or from the average of both, has a negligible effect on this analysis (see Figs. S5 to S46 in the supplement): the mean temperature profiles measured by the two beams differ little in shape and absolute temperature.

The resulting daily mean temperature profile from the chosen beam is then weighted with a Gaussian function. To identify the parameters of best agreement between the two instruments, the weighting needs to be performed with different centre altitudes and FWHM of the Gaussian weighting function. I choose 40 centre altitudes between 81.8 km and 92.8 km, and 40 FWHM between 4.7 km and 15.7 km. The bins are separated by 282 m, which is twice the altitude resolution of the lidar at a zenith angle of 20°.

For each centre altitude $z$ and full width at half maximum $d$, the weighted nightly mean temperature $\overline{T}$ is given by

$$\overline{T}_{z,d} = \sum_{h=0}^{N} f_h T_h, \tag{1}$$

where $f_h$ is the weighting factor at each altitude $h$, corresponding to the choice of $z$ and $d$, between the lowest and highest useable altitude. The temperature at a given altitude $h$ in the nightly mean temperature profile is denoted by $T_h$.

The nightly mean temperatures could have been weighted with their corresponding uncertainty, but this is not how the nightly mean temperature is calculated from spectrom-

eter data. Besides, the assumption of a Gaussian distribution is a simplification. For example, it is possible that the real OH\* layer has a shape best approximated by a skewed or a multi–peak Gaussian distribution. Individual OH\* profiles measured by sounding rockets (Baker and Stair Jr., 1988) generally do not have a Gaussian shape, while average profiles calculated from rather short observation intervals (Noll et al., 2015) can be well–approximated by a Gaussian distribution (Noll et al., 2016, Fig. 8). The GRIPS 9 data were taken with a one–minute resolution, from which I computed a nightly mean temperature. This is similar to the data shown in Noll et al. (2016), justifying this simplification of a Gaussian distribution. The effect of using different weighting functions, though all being approximately Gaussian, to compute an OH\*(3–1)–equivalent temperature has been shown to be smaller than 3 K for any of the functions considered (French and Mulligan, 2010). For nightly mean data, which I consider here, there does not seem to be any evidence for a distribution substantially different from a Gaussian shape. See Sect. 4 of the supplement to this article for a digression on this topic.

## 3   Results and discussion

The nightly mean temperatures, measured by the Na lidar at ALOMAR, for four aribtrary nights (see the supplement for full data set) are shown in Fig. 1. The nightly mean OH\*(3–1) rotational temperature measured by GRIPS 9 is also shown, but note that I have assumed this temperature as representative of the OH\*(3–1) layer at 87.4 km with a full width at half maximum of 8.4 km.

To determine how well the temperatures, measured independently by the two instruments, actually agree, I compute the nightly mean Gaussian–weighted Na lidar temperature from the temperature profiles shown in Fig. 1 for 40 different centre altitudes and 40 different FWHM. To find the altitude of the best agreement, I calculate the absolute temperature difference between GRIPS 9 and the OH\*(3–1)–equivalent temperatures calculated from the Na lidar data. This allows one, in principle, to find the centre altitude and full width at half maximum where the temperature difference is smallest, that is, where the agreement is best.

One may argue that this analysis is too general, because I have also chosen centre altitudes or full widths at half maximum which seldom or never occur in reality, for instance. While this is probably true, at least to some extent, restricting the analysis to narrower layer widths and fewer centre altitudes would only change a numerical result, if I were to compute a mean centre altitude. However, as will be evident, it is not advisable to compute such statistics from these data because of the inherent ambiguity. Hypothetically, in case the measured temperature profile were not the true temperature profile, but offset from the truth by a certain value, the effect on the results would be similar to that of a different set of Einstein coefficients: any ambiguity would remain, only

the values of the altitude– and width–dependent temperature difference would change.

The ambiguity is visible in Fig. 2, which shows the difference of the nightly mean temperatures for the different centre altitudes and FWHM for four nights. The temperature difference is given by

$$\triangle T(z,d) = T_{\mathrm{OH^*(3-1)}} - \overline{T}_{z,d}, \tag{2}$$

where $z$ is the centre altitude of the Gaussian weighting function, and $d$ is its full width at half maximum. Positive temperature differences thus indicate that the temperatures measured by GRIPS 9 are warmer than the weighted temperatures measured by the Na lidar.

Figure 2 shows that there is, in most cases, more than one combination of centre altitude and full width at half maximum that yield the smallest temperature difference— regardless of the measurement duration. A temperature agreement of $|\triangle T| < 2$ K can often be obtained from several combinations of centre altitude and full width at half maximum. Even with the width fixed at, say, 8.4 km, there are nights on which the altitude determination is ambiguous, see Fig. 2(c): the nightly mean altitude may be either $\sim$84 km or $\sim$90 km. Both are realistic values (Winick et al., 2009). Even if the width of the OH\*(3–1) layer is taken into account, one cannot determine its altitude unambiguously from the temperature measurements by lidar. However, not taking the FWHM into account might give false confidence in a so–determined altitude, because the ambiguity might be invisible.

Temperature differences between the two instruments of $|T| \geq 10$ K can arise simply from assuming a certain fixed width of the OH\* layer. On a few nights, the mean temperatures differ by more than 10 K for all sensible combinations of centre altitude and full width at half maximum, see Fig. 2(b). Thus, I argue that a systematic temperature offset between the instruments cannot be detected beyond doubt, and that apparent offsets are not necessarily caused by real systematic effects.

Other notable observations are that the temperature agreement can be almost independent of the chosen parameters, see Fig. 2(b), and that, generally, the agreement is not best at the mean centre altitude of 87.4 km and the mean full width at half maximum of 8.4 km (Baker and Stair Jr., 1988). Still, the results do not undermine the assumption that the OH\* rotational temperature can be used as an estimate of the nightly mean temperature at 87 km (She and Lowe, 1998).

In an ideal case, an emission profile is available for the interpretation of ground–based rotational temperature measurements. However, this is rather seldom the case. Whenever the emission altitude and width of an airglow layer is not readily available, a stationary altitude and width (typically, $\sim$87 km and 8 km, respectively) must be assumed in the interpretation of ground–based rotational temperatures. In cases where no measurement of the actual altitude and width is available, a comparison like this one can be a second–best

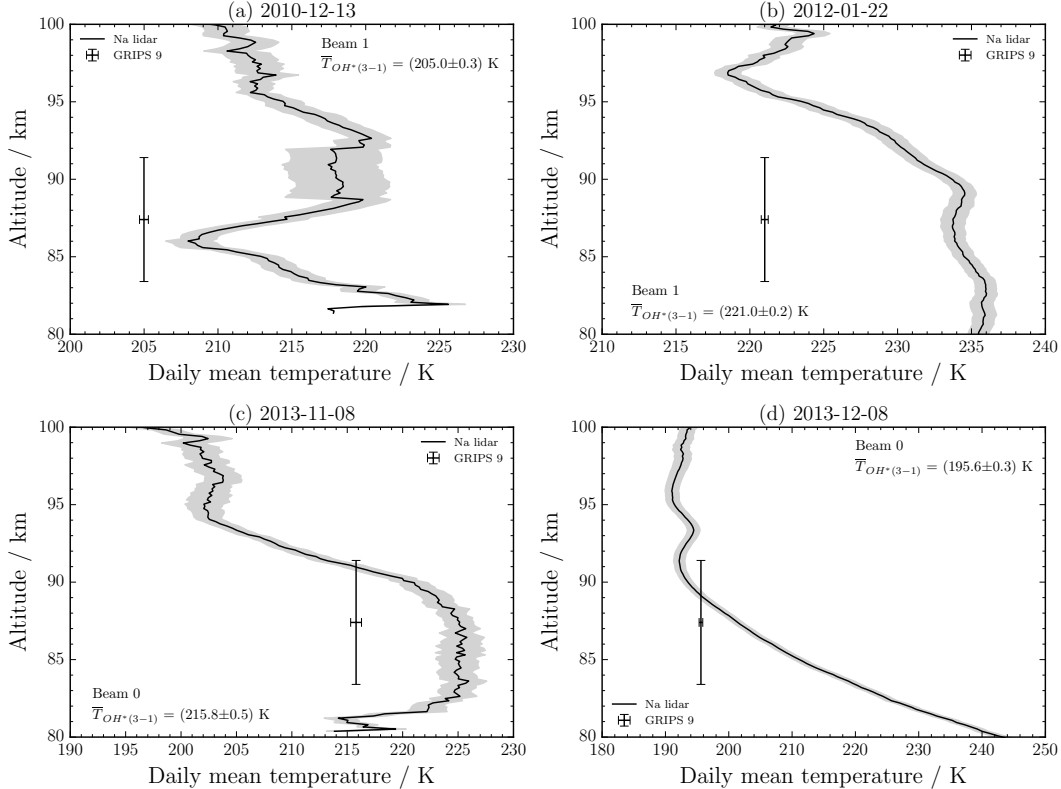

**Figure 1.** Nightly mean temperature profiles and standard error (grey–shaded areas) measured by the Na lidar at ALOMAR. Also shown is the mean temperature and the standard error measured by GRIPS 9 during the same period of time. This temperature is plotted at an altitude of 87.4 km, and the error bars are meant to indicate a full width at half maximum of 8.4 km. Date and common measurement duration: (a) 13 December 2010, 4:22 h; (b) 22 January 2012, 13:46 h; (c) 8 November 2013, 3:10 h; (d) 8 December 2013, 14:17 h.

option, despite the ambiguity in the altitude determination: assuming a stationary altitude of 87.4 km and a width of 8.4 km for all 42 nights, I compute the temperature difference for each of the nights under these assumptions. The results then indicate how representative the OH*(3−−1) temperature is of this altitude and width. Note that this temperature difference is not a measurement uncertainty or error.

Figure 3 shows the temperature difference $\triangle T$ for each of the 42 nights, assuming a fixed centre altitude of 87.4 km and a fixed FWHM of 8.4 km. It is thus possible to quantify how representative this proxy is of 87 km (She and Lowe, 1998) and this width. The maximum and minimum temperature differences are 12 K and –20 K (Fig. 3). Because the temperature differences shown in Fig. 3 are not normally–distributed, the mean (and corresponding standard deviation) of the temperature difference is not a sensible choice. A more appropriate measure in this case is half the difference between the maximum and minimum temperature difference, yielding ±16 K, which is specific to this comparison and may be different for other dates, locations, and instruments. I obtain similar values for other combinations of altitude and width. What this quantity implies is that, on any given day, the OH*(3–1) layer may be higher or lower than 87.4 km,

or that its thickness is not 8.4 km, or that its shape is not Gaussian, or any combination of these. Although ±16 K may seem large, it does not seem too bad to me for two reasons. First, I do not have available any actual information about the OH*(3–1) layer's shape and altitude, I just assumed an altitude and width I thought was probable. Second, in the mesopause region, temperature differences of ±16 K may arise from altitude variations of about 3 km, assuming an adiabatic lapse rate of ∼5 K km$^{-1}$; that is, if the OH*(3–1) layer were at about 84 km or 90 km while I have assumed it to be at about 87 km. This argument seems more valid for individual days, though, and such altitude variations do occur (Teiser and von Savigny, 2017, Figs. 5 and 8). Keep in mind that this is not a measurement uncertainty. Also, this is a result specific to this comparison.

Even though it may seem tempting to compute the mean centre altitude and FWHM of the OH*(3–1) layer based on this spectrometer—lidar comparison, such a result would be misleading and camouflage the ambiguity shown in Fig. 2. The main reason for this ambiguity is the nightly mean temperature profile. To unambiguously determine the altitude of, for example, the OH*(3–1) layer from a comparison with weighted temperatures measured by lidar, the nightly

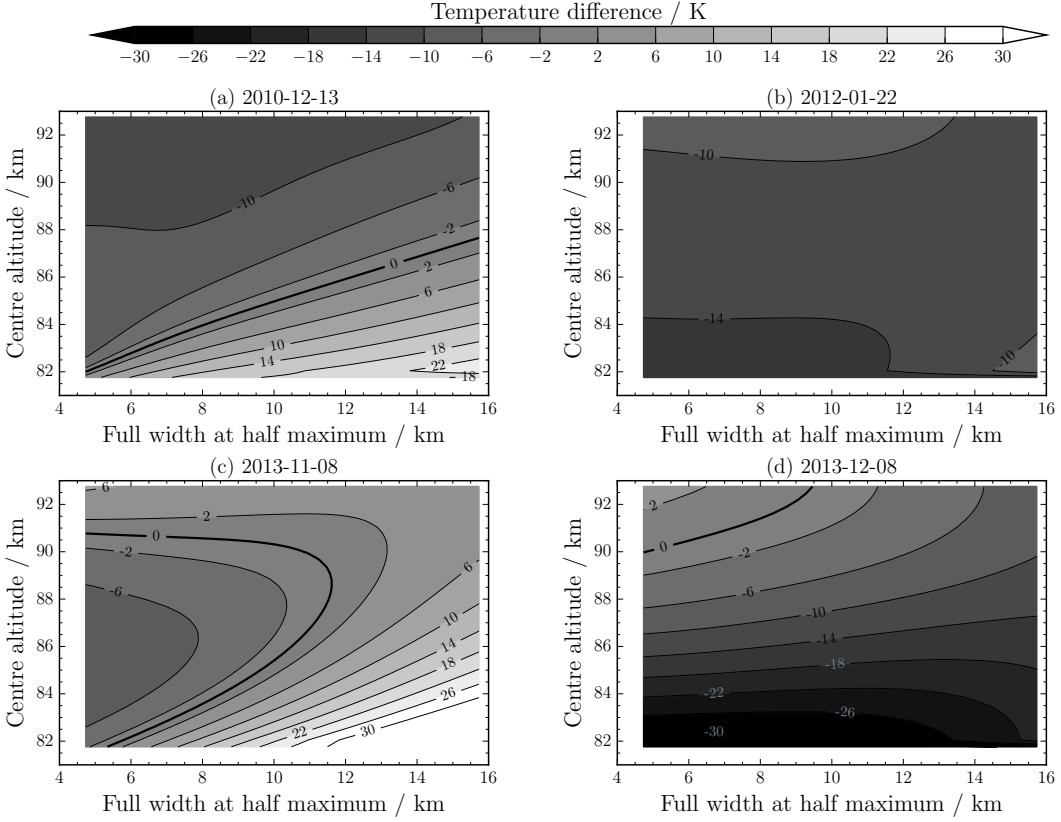

**Figure 2.** Nightly mean temperature difference on 13 December 2010 between GRIPS 9 and artifical OH\*(3–1) temperatures from the Na lidar at ALOMAR. Temperature difference is shown as $\triangle T(z,d) = T_{\mathrm{OH}^*(3-1)} - \overline{T}_{z,d}$, and is a function of chosen centre altitude and full width at half maximum. Weighted temperatures are calculated despite possibly missing data at the the boundaries. Positive values indicate that GRIPS 9 measured an apparent warmer temperature than observed by the lidar for the given parameters. The 0–contour line indicates exact agreeement. Measurement durations are given in Fig. 1.

mean temperature profile throughout the layer would have to show certain characteristics. A monotonously increasing or decreasing temperature profile is not enough. One could then vary the assumed layer width of the Gaussian arbitrarily without changing the temperature agreement. A strictly monotonous temperature profile is seldom the case (see also the mean temperature profiles in the supplement). Gravity waves, tides, and turbulence drive the temperature profile away from a perfect adiabatic lapse rate, for example. Mesospheric inversion layers can persist for several hours (Szewczyk et al., 2013), and can be a prominent feature even after averaging. Thus, measurements shorter than a few hours are not necessarily the cause of the ambiguity.

An important systematic uncertainty of the method used here is the uncertainty of the absolute OH\*(3–1) rotational temperature due to the set of Einstein coefficients used in its computation (C. Schmidt, pers. comm.). The effect of a different set of Einstein coefficients is that the best agreement would then appear at a different centre altitude and FWHM. Despite being an uncertainty, it further corroborates the re-

sults, namely that it is not possible to determine unambiguously the altitude and width from lidar measurements alone.

## 4  Conclusions

I compared nightly mean temperatures from 42 coincident measurements of the Na lidar at ALOMAR and GRIPS 9, covering the period from October 2010 to April 2014. To approximate the OH\* layer temperatures measured by GRIPS 9, I weighted the lidar temperatures using Gaussian functions with 40 different centre altitudes and 40 full widths at half maximum.

To interpret variations seen in OH\* rotational temperatures, be it on a decadal or hourly scale, the emission altitude must be known. A climatological OH\*(3–1) layer emission altitude of 87 km is not incompatible with this study, but the nightly OH\* layer height generally cannot be determined unambiguously from temperature measurements by lidars, regardless of whether a variable layer width is taken into account. This is probably also true for any other in-

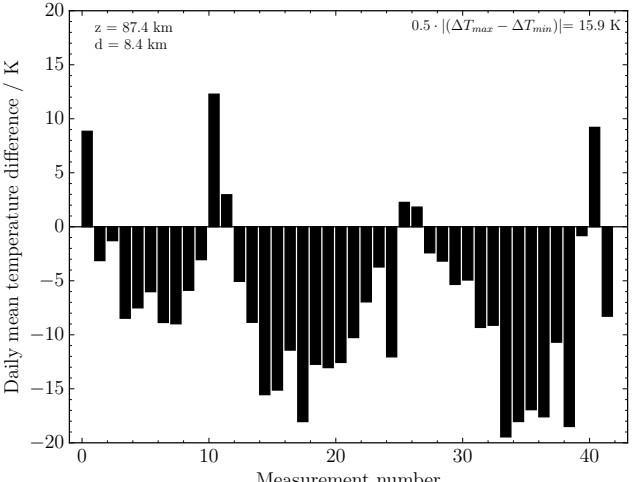

**Figure 3.** Temperature difference for each day, assuming a fixed a centre altitude of 87.4 km and a full width at half maximum of 8.4 km for each day. See the supplement for a list of all measurements.

strument that measures temperature profiles, and is true for any location. Because different combinations of centre altitude and full width at half maximum often yield very good temperature agreement, the parameters of the approximated OH*(3–1) layer are usually ambiguous. Still, for any such analysis, the width of the Gaussian should not be fixed at some value, because a fixed width (a) is incompatible with observations (Baker and Stair Jr., 1988), and (b) gives false confidence in the altitude determination because the ambiguity may often be invisible. To determine the emission altitude and width of any—not just of the OH*(3–1) transition—of the airglow layers unambiguously, satellite– or rocket–based observations of the volume emission rate profiles are necessary (Mulligan et al., 2009), for example. Even though these measurements do not usually coincide with the spectrometer measurements, a miss distance of up to 500 km or a miss time of several hours can well be accepted (French and Mulligan, 2010).

In case no altitude and width measurements of an airglow layer are available for the interpretation of ground–based rotational temperature measurements, it is possible to estimate from a comparison with lidars (or similar instruments) how representative a measured temperature is of an assumed altitude and width. In the present analysis, I assumed a stationary altitude of 87.4 km and width of 8.4 km. On average, I found the OH*(3–1) rotational temperature to be representative to within $\pm 16$K of the temperature at this altitude and width. This figure is specific to this comparison, and is not a measurement uncertainty.

## 5   Code availability

I will provide the code upon request.

## 6   Data availability

The Na lidar data used in this article are archived at Harvard Dataverse (Dunker, 2018). The airglow observations of GRIPS 9 are archived at the World Data Center for Remote Sensing of the Atmosphere (WDC–RSAT, https://wdc.dlr.de).

*Competing interests.* There are no competing interests.

*Acknowledgements.* I gratefully acknowledge the valuable collaboration and discussions with Carsten Schmidt of DLR. I am grateful to Michael Bittner of DLR for making possible my fruitful working visits with his research group, and to Sabine Wüst, also of DLR, for making available the online platform for GRIPS data. Numerous discussions with Ulf–Peter Hoppe and Dallas Murphy's comments greatly improved this manuscript. I thank Katrina Bossert and Bifford P. Williams, GATS, Inc., as well as the ALOMAR staff, for conducting several of the lidar measurements. The Research Council of Norway funded this work under grant 216870/F50, as well as many Na lidar measurements under grant 208020/F50. The Na lidar at ALOMAR is a National Science Foundation Upper Atmosphere Facility instrument, funded under grant NSF AGS–1136269. The airglow observations at ALOMAR were partly funded by the Bavarian State Ministry for Environment and Consumer Protection (project BHEA, grant number TLK01U–49580). The publication charges for this article were funded by a grant from the publication fund of UiT The Arctic University of Norway.

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
