# Peer review of "The airglow layer emission altitude cannot be determined unambiguously from temperature comparison with lidars"

_Atmospheric Chemistry and Physics, 2017_

## Referee Comment (RC1) · Anonymous Referee #1 · 22 Oct 2017

This paper deals with the possibilities and limitations of co-analyzing OH Meinel night-glow and sodium resonance lidar in terms of mesospheric temperatures. Ground-based OH Meinel spectroscopy has long been an important tool for monitoring temperatures in the upper mesosphere. Question about the height, width and variability of the OH emission layer are decisive for the interpretation of resulting temperature time series. Direct comparison with co-located sodium resonance lidar can provide some constrains on the geometry of the emission layer. However, as the current paper demonstrates, the knowledge accessible about the detailed layer geometry remains limited. In particular, a simultaneous and independent assessment of the two parameters layer altitude and layer width is generally not possible.

[Figure]

This comparison study is not the first of its kind. The author correctly refers to the earlier work by von Zahn et al. (1985) and Pautet et al. (2014). Nevertheless, the data presented here provide a valuable contribution to assessing the limitations of OH nightglow temperature studies. As expressed in the title, the author suggests that the major conclusion of the paper is that the height of the OH emission layer cannot unambiguously be determined by this kind of study. I am not sure, however, that this really is the most interesting message from this work to the atmospheric community. I actually think that the most important result for the community is the quantification of the uncertainty of the OH temperature measurements, as shown e.g. in Figure 2. Therefore, I would like to suggest to the author to consider shifting the major message of the paper towards the actual temperature uncertainty.

My major concern with this paper is the discussion of the ambiguity between OH layer altitude and layer width. The interpretation of the data as shown e.g. in Figure 1 is not convincing and should be modified. In my opinion, this requires a major revision, after which the paper can be regarded as an interesting contribution to ACP.

Starting point of the study are coincident detailed temperature profiles with the Na lidar and "column" OH temperatures determined by ground-based Meinel spectroscopy. The basic analysis idea is then to infer a geometry of the OH layer based on the requirement that the Na temperature integrated over the OH layer must be consistent with the spectroscopic OH temperature. My concern is that the author intends to determine two unknowns (layer height and layer thickness) while there is only one observational constraint (the difference between OH temperature and integrated Na temperature). The problem is thus under-determined. This means that the statement given in the title is rather trivial: the under-determined problem will not allow for an unambiguous retrieval of the two independent parameters.

The author describes the mathematical problem as a "minimization": By minimizing the difference between OH temperature and integrated Na temperature, the two unknowns layer altitude and layer width should be fitted. This approach would be appropriate

for an overdetermined system. For this underdetermined system, however, there is nothing to be minimized. On the contrary, for typical conditions there is an infinite number of combinations of OH layer altitude and layer thickness that result in exact zero difference between OH temperature and integrated Na temperature. This is obvious from Figure 1: with the exception of 2012-01-22, all plots feature a zero contour line that represents the pairs of layer altitude and thickness that generate exact solutions. On page 5, lines 11-12, the author refers to "more than one combination [...] that yield the smallest temperature difference". Why should one talk about smallest differences when there are obviously exact solutions that yield exact zero?

Hence, I urge the author to rethink the overall analysis concept and the mathematical formulation. As pointed out above, I would actually recommend the author to shift focus from the somewhat trivial question of "OH layer ambiguity" to the more interesting question of "OH temperature uncertainty".

Some other specific comments:

page 2, lines 27-29: Some basic description of the iterative procedure for temperature retrieval should be provided. Simply referring to the (hard to access) Ph.D. thesis by Lange (1982) makes it very difficult to follow the paper.

page 5, lines19-22: It is stated that for some nights reasonable solution cannot be found. I wonder whether this in part is a consequence of strong variations of temperature and/or OH layer geometry within a given night. This would make the use of nightly means problematic. To check this, it might be instructive to break up the analysis into several time intervals for a given night.

page 5, lines 29-30: It is stated that the temperature differences are not normally distributed, and that for this reason it is not possible to determine mean and standard deviation. Therefore, the author suggests to use half the difference between maximum and minimum temperature difference as a measure for uncertainty, which results in a rather large range of +/- 16 K. Even though the differences are not normally distributed,

I would still argue that it might be more meaningful to talk about a "mean error", which would result in a significantly smaller measure of the uncertainty than +/- 16 K. On page 7, lines 17-20, the author uses a different notation, talking about a temperature proxy being "representative within +/- 16 K", thus referring to this interval more as "outer limits" than a conventional uncertainty interval (mean error). Please make sure to give the reader a clear feeling for what is meant by the "uncertainty interval" +/- 16 K.
* * *

---

## Referee Comment (RC2) · Anonymous Referee #2 · 13 Nov 2017

The paper addresses the question to what extent temperature observations taken by a Na-LIDAR-instrument can be used to help estimating the height and the width of the hydroxyl airglow layer. The author uses measurements of the OH*-rotational/vibrational temperatures which were derived by the GRIPS-9 instrument at ALOMAR, Norway, by the German Aerospace Center, DLR. Temperature observations at the upper mesosphere / lower thermosphere are of interest to a broad variety of scientific questions ranging from temperature trend studies to atmospheric circulation aspects as well as to studies dedicated to detailed process understanding. The manuscript therefore falls well into the scope of ACP. The authors conclude that lidar measurements cannot clearly determine the OH emission height and width of the OH-layer. The manuscript

is scientifically sufficiently sound to be published as is.

---

## Author Comment (AC1) · 6 Jan 2018

Dear colleagues

Thank you very much for your comments and recommendations. My answers are below and the a list of proposed changes is at the end of this reply.

Kind regards
Tim Dunker

**Point–by–point reply to reviewer 1**

**This comparison study is not the first of its kind. The author correctly refers to the earlier work by von Zahn et al. (1985) and Pautet et al. (2014). Nevertheless, the data presented here provide a valuable contribution to assessing the limitations of OH nightglow temperature studies. As expressed in the title, the author suggests that the major conclusion of the paper is that the height of the OH emission layer cannot unambiguously be determined by this kind of study. I am not sure, however, that this really is the most interesting message from this work to the atmospheric community. I actually think that the most important result for the community is the quantification of the uncertainty of the OH temperature measurements, as shown e.g. in Figure 2. Therefore, I would like to suggest to the author to consider shifting the major message of the paper towards the actual temperature uncertainty.**

Answer:
First of all, I want to point out that Fig. 2 does not show measurement uncertainties of the OH* rotational temperature measurements. I will make this clear in the figure caption. What the figure shows is the following: if we measure the OH* rotational temperature, but are ignorant of the actual height and width of the OH* layer—as is often the case—and thus assume the layer to be at 87.4 km and being 8.4 km wide, then the measured rotational temperature is representative of this altitude and width to within $\pm 16$ K. This is not a classic measurement uncertainty of any of the two instruments.
I will make this point much clearer in the revised manuscript. Also, I am hesitant to shift the message as suggested, because I am not in a good position to analyze the actual OH rotational temperature uncertainties. Please see also my reply to the specific point regarding page 5, lines 29 to 30.

**My major concern with this paper is the discussion of the ambiguity between OH layer altitude and layer width. The interpretation of the data as shown e.g. in Figure 1 is not convincing and should be modified. In my opinion, this requires a major revision, after which the paper can be regarded as an interesting contribution to ACP. Starting point of the study are coincident detailed temperature profiles with the Na lidar and "column" OH temperatures determined by ground-based Meinel spectroscopy. The basic analysis idea is then to infer a geometry of the OH layer based on the re- quirement that the Na temperature integrated over the OH layer must be consistent with the spectroscopic OH temperature. My concern is that the author intends to determine two unknowns (layer height and layer thickness) while there is only one observational constraint (the difference between OH temperature and integrated Na temperature). The problem is thus under–determined. This means that the statement given in the title is rather trivial: the under–determined problem will not allow for an unambiguous retrieval of the two independent parameters. The author describes the mathematical problem as a "minimization": By minimizing the difference between OH temperature and integrated Na temperature, the two unknowns layer altitude and layer width should be fitted. This approach would be appropriate for an overdetermined system. For this underdetermined system, however, there is nothing to be minimized. On the contrary, for typical conditions there is an infinite number of combinations of OH layer altitude and layer thickness that result in exact zero difference between OH temperature and integrated Na temperature. This is obvious from Figure 1: with the exception of 2012–01–22, all plots feature a zero contour line that represents the pairs of layer altitude and thickness that generate exact solutions. On**

**page 5, lines 11-12, the author refers to "more than one combination [...] that yield the smallest temperature difference". Why should one talk about smallest differences when there are obviously exact solutions that yield exact zero? Hence, I urge the author to rethink the overall analysis concept and the mathematical formulation. As pointed out above, I would actually recommend the author to shift focus from the somewhat trivial question of "OH layer ambiguity" to the more interesting question of "OH temperature uncertainty".**

Answer:

The reviewer's major concern is the discussion of the ambiguity between OH* layer altitude and layer width. The reviewer suggests that the interpretation of the data such as shown in Figure 1 must be modified as a major revision. It is precisely the mentioned ambiguity that is the main subject of this manuscript. The interpretation of the data such as shown in Figure 1 was the subject of many discussions such as mentioned in the acknowledgement. The reviewer may be right that it is trivial that the combination layer height and layer thickness cannot be unambiguously determined, however: (a) the papers that I cite tried to do exactly that, (b) even if we assume one of the parameters to be stationary, the ambiguity remains unresolved.

I disagree that "...there is only one observational constraint (the difference between OH temperature and integrated Na temperature)". I suspect this is a misunderstanding. On the one hand, we have one temperature value from the airglow measurement. The literature that I cite assumes that the temperature value is a weighted mean of an (typically unknown) intensity profile. On the other hand, we have a temperature profile measured at the same place in the sky and during the same time interval with the lidar. The temperature profiles are shown in the supplement, but I realize that it is much better to add a figure with the temperature profiles for the days shown in the manuscript itself. To clarify, I will add the information in the first paragraph of section 2 that the lidar measures $N = 300$ altitude bins (150 m) of temperature between 70 km and 115 km. Therefore, we have $N+1$ measured values to determine the two parameters OH*layer height and OH* layer width. The system appears to be overdetermined.

The reviewer points out correctly that "...all plots (with one exception) feature a zero contour line that represents the pairs of layer altitude and thickness that generate exact solutions". The existence of such a contour line is an important result of this paper. Before I generated these plots, many colleagues thought that all plots like the four in Fig. 1 would display a minimum at one combination of altitude and width, looking like a valley in a map with altitude contours. If the existence of such a contour line was not known and not published before, I think it is not as trivial a fact as the reviewer maintains.

I thank the reviewer for the recommending to shift towards the "OH temperature uncertainty". Several authors, who measured such rotational temperatures, calculated the uncertainties of the measurements (e.g., Bittner et al., 2000; Sigernes et al., 2003; Schmidt et al., 2013). My work described in this manuscript concerns the uncertainty of the height where OH* airglow temperatures are measured. The data show that this height is not a constant over time, which is in good agreement from what is known from rocket– and satellite–based measurements. The figure ought not to be interpreted as a temperature measurement uncertainty, and I would like to keep the focus as is, but instead expand on the interpretation of this figure.

**Some other specific comments:**
**page 2, lines 27-29: Some basic description of the iterative procedure for temperature retrieval should be provided. Simply referring to the (hard to access) Ph.D. thesis by Lange (1982) makes it very difficult to follow the paper.**

Answer:

The data analysis is exactly as described in Schmidt et al. (2013, Sect. 3.3). I will replace the citation of Lange (1982) with the much better, and more useful, citation of Schmidt et al. (2013).

**page 5, lines 19-22: It is stated that for some nights reasonable solution cannot be found. I wonder whether this in part is a consequence of strong variations of tempera- ture and/or OH layer geometry within a given night. This would make the use of nightly means problematic. To check**

**this, it might be instructive to break up the analysis into several time intervals for a given night.**

Answer:
Yes, I think it is very reasonable to assume that also the OH* layer's height and width varies during the course of a night. This, at least, is true of the Na layer, and I do not see why the OH* should not exhibit such variation. On page 5, lines 32 to 33, I write that "on any given day, the OH*(3–1) layer is higher or lower than 87.4 km, or that its thickness is not 87.4 km, or that its shape is not Gaussian, or any combination of these." I do not want to show such an analysis here, because the manuscript's main point is the ambiguity. It is possible to gain some insight from a detailed analysis of all one–minute mean temperatures (and the lidar's temperature profiles), but the message of the manuscript remains the same. It would also probably open up a third thread, which does not seem essential to me. I will point out, though, that a detailed analysis might yield insight into the variability.

**page 5, lines 29-30: It is stated that the temperature differences are not normally distributed, and that for this reason it is not possible to determine mean and standard deviation. Therefore, the author suggests to use half the difference between maximum and minimum temperature difference as a measure for uncertainty, which results in a rather large range of +/- 16 K. Even though the differences are not normally distributed, I would still argue that it might be more meaningful to talk about a "mean error", which would result in a significantly smaller measure of the uncertainty than +/- 16 K. On page 7, lines 17-20, the author uses a different notation, talking about a temperature proxy being "representative within +/- 16 K", thus referring to this interval more as "outer limits" than a conventional uncertainty interval (mean error). Please make sure to give the reader a clear feeling for what is meant by the "uncertainty interval" +/- 16 K.**

Answer:
I must emphasize that this figure is not an uncertainty—I have therefore avoided to speak of an "uncertainty" anywhere in the manuscript in connection with this figure, because it should not be understood as such. I will write this more explicitly in the revised manuscript. The figure is neither a statistical nor systematic uncertainty of the OH* rotational temperature. The reviewer's interpretation of this figure as "outer limits" seems very sensible to me. Very often, there is no altitude and width information available for ground–based measurements of OH* rotational temperatures, but for the rotational temperature to have any meaning, we must assume it to be representative of the temperature at some altitude. A typical choice has been 87 km. This is where the lidar–spectrometer comparison is useful: we do not have any information on the actual shape and height of the OH* layer, but the temperature comparison can be done under the assumption of a static altitude (87.4 km) and a static width (8.4 km). Now, for once, this result is actually specific to this dataset: the daily temperature difference using these assumptions then can be used to assess how well the measured temperature is representative of this height and this width. One might choose other altitude and width, and I will do this in more detail in the revision.
The value of ±16 K is not too bad an "outer limit" of represantativeness, in my opinion. If we think of it in terms of a possible altitude variation, using the adiabatic lapse rate of the mesopause region, these ±16 K amount to less than ±2 km in altitude (neglecting the width) for any given day. Such variations do occur frequently (Teiser and von Savigny, 2017, Figs. 5 and 8). I try to avoid using the terms error or standard deviation. The discussion would be the same, though, but there is the danger of clouding even more that I am not dealing with measurement uncertainties of any of the two instruments. On page 5, line 32, I speak of an "offset". I now think that this term is not a good choice, and probably makes the interpretation harder than necessary. In the revised manuscript, I will replace the term "offset". I will also point out that using this value of ±16 K is (a) specific for this study and may be different at other locations and for other instruments, and (b) that its use is only the second–best option—in my opinion, it is best to acquire knowledge of the actual OH* intensity profile.

**List of proposed changes**

I have included in this list a few changes that were not requested by the reviewers. These changes concern a typographical error and a few clarifications. I will consider all aspects of the foregoing discussion, of course, even it does not appear in this preliminary list below.

- I will add a figure with four panels, showing nightly mean temperature profiles measured by the lidar and the OH*(3–1) rotational temperature measured by GRIPS 9. Each of the panels will show data corresponding to the dates shown the manuscript. The new figure will become Fig. 1, and the "old" Fig. 1 will become Fig. 2. What is now Fig. 2 will become Fig. 3.

- In Sect. 2, I will add several sentences to clarify that the "determination" of altitude and width is based on many temperatures measured by the lidar.

- I will expand the discussion and will clarify what is meant by the value of 16 K, that it is not a measurement uncertainty, following the discussion during the review. I will also point out that it is a result specific for this study, while the other conclusions are general. Furthermore, I will make it clear that the best option is to always obtain an OH* intensity profile, and that the second–best option is to assume an altitude for the measured temperature.

- Though not the aim or topic of the manuscript, I will point out that an analysis of the data at shorter time resolution might yield insight into the short–term variability.

- I will replace the citation of Lange (1982) with Schmidt et al. (2013, Sect. 3.3)

- I will investigate how the choice of a different altitude and width impact the temperature difference between the instruments.

- p. 2, l. 12: "of the applied Gaussian"

- p. 2, l. 15: "assuming a stationary width"

- p. 2, l. 23: "are based on measurements with a temporal resolution of one minute."

- p. 3, ll. 15–16: delete sentence "The assumptions..., though."

- p. 3, l. 34: "substantially"

- p. 4, l. 4: " This allows one to find..."

- p. 5, ll. 11-12: Remove "Rather"

- I will consider changing the title: right now, it appears as if I were suggesting the conclusion only applies to this specific nightglow layer. I would rather argue that it applies to any nightglow layer.

**References**

Bittner, M., D. Offermann, and H. H. Graef (2000). Mesopause temperature variability above a midlatitude station in Europe. *J. Geophys. Res.–Atmos.* 105(D2), pp. 2045–2058. DOI: 10.1029/1999JD900307.

Lange, G. (1982). *Messung der Infrarotemission von OH\* und O2($^1\triangle_G$) in der Mesosphäre*. Dissertation, WUB–DI 82–3. In German. Gesamthochschule Wuppertal.

Schmidt, C., K. Höppner, and M. Bittner (2013). A ground–based spectrometer equipped with an InGaAs array for routine observations of OH(3–1) rotational temperatures in the mesopause region. *J. Atmos. Sol.–Terr. Phys.* 102, pp. 125–139. DOI: 10.1016/j.jastp.2013.05.001.

Sigernes, F., N. Shumilov, C. S. Deehr, K. P. Nielsen, T. Svenøe, and O. Havnes (2003). Hydroxyl rotational temperature record from the auroral station in Adventdalen, Svalbard (78°N, 15°E). *J. Geophys. Res.– Space* 109(A9), 1342. DOI: 10.1029/2001JA009023.

Teiser, G. and C. von Savigny (2017). Variability of OH(3–1) and OH(6–2) emission altitude and volume emission rate from 2003 to 2011. *J. Atmos. Sol.–Terr. Phys.* 161, pp. 28–42. DOI: 10.1016/j.jastp.2017.04.010.

---

## Author Response (AR1)

Dear colleagues

I sincerely thank you again for agreeing to review my manuscript and for your comments. All changes are marked in the manuscript. I hope I have implemented your comments satisfactorily.

Kind regards
Tim Dunker

[revised manuscript text omitted]

---

## Author Response (AR2)

**List of changes**

[revised manuscript text omitted]